# First-Step Advantage: Importance of Starting Right in Multi-Step Math Reasoning

## Abstract

Language models can solve complex reasoning tasks better by learning to generate rationales for their predictions. Often these models know how to solve a task but their auto-regressive decoding nature leads to incorrect results if they start incorrectly. We observe that smaller models in particular when corrected, can solve a task that they would have otherwise struggled with. We demonstrate this phenomenon by using a larger model to guide smaller models, which leads to significantly improved performance (up to +24 points on the GSM8K dataset by 7B models). To assist smaller models in initiating the starting step, we propose `QuestCoT`, where a smaller model first *asks itself how to start*, before proceeding with a chain of reasoning. On various multistep mathematical reasoning datasets over multiple smaller models, we show that getting the right start can lead to significant performance gains across all models (gains of up to +6 points on GSM8K, +9 on SVAMP, +5 on ASDiv, and +7 on MultiArith).

## 1 Introduction

Over the years, large language models (LLMs) have improved their reasoning abilities by explaining their intermediate thoughts Wei et al. (2022). This trend has been extended to smaller models [1], either through pre-training Jiang et al. (2023); Magnusson et al. (2023), fine-tuning Yu et al. (2023); Shao et al. (2024), or knowledge distillation Shridhar et al. (2023b); Yuan et al. (2023); Magister et al. (2023); Hsieh et al. (2023). While it is commonly assumed that smaller models acquire new knowledge through fine-tuning or distillation, recent research by Gekhman et al. (2024) suggests that the acquisition of new knowledge is quite slow. Instead, models often improve in the areas they are already familiar with. This suggests that while models may have the knowledge to solve a given task, they struggle to understand how to apply it effectively.

Wang et al. (2023b) demonstrates that model accuracy improves significantly when multiple reasoning chains are generated, indicating that the model understands how to answer the given problem. However, models often struggle to select the correct initial chain, and if they start on an incorrect reasoning path, it becomes difficult to fix it due to the autoregressive nature of decoding. Similarly, in our work, we observed that if a smaller model initiates an incorrect reasoning chain, it will continue down that incorrect path. Conversely, if the initial step is correctly determined, the model can successfully complete tasks that it would otherwise find challenging.

In this work, we first investigate whether providing initial guidance can improve the reasoning capabilities of smaller language models. We then investigate whether the quality of this initial guidance varies depending on the expertise of different large language models (LLMs). In particular, we investigate whether smaller models can use this guidance without fine-tuning or additional training, and whether models of different sizes benefit equally. Finally, we investigate whether the benefits of initial guidance extend beyond simple two-step problems to tasks requiring four to eight steps of reasoning.

---

[1] we use smaller models in a relative sense and most of our experiments are carried out on models smaller or equal to 7B parameters

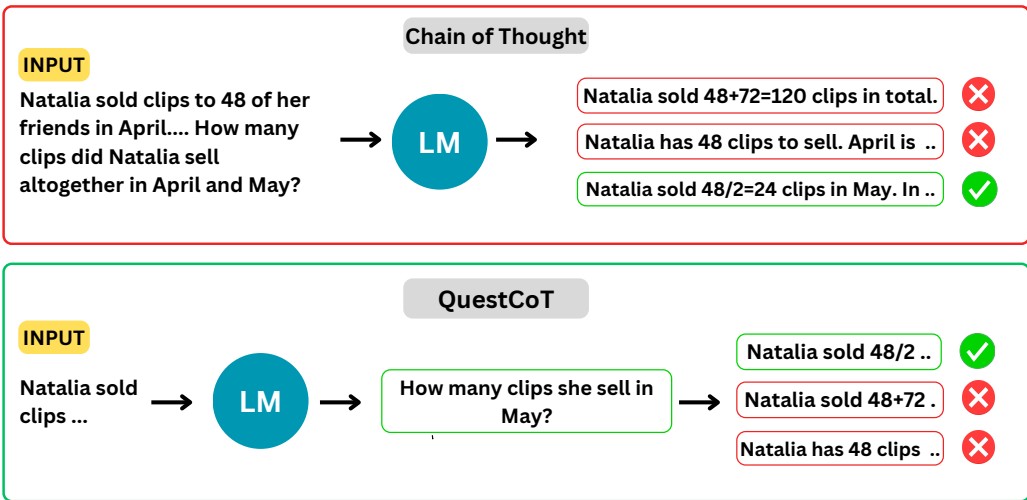

Figure 1: **Comparison between Chain-of-Thought (CoT) approach and `QuestCoT`**. The CoT approach enables a Language Model (LM) to generate accurate answers through multiple samplings, yet it frequently struggles to confidently select the correct one. Conversely, `QuestCoT` utilizes self-question-guided generation, which facilitates the model's ability to choose the appropriate reasoning chain with higher confidence.

Once the critical role of initial step guidance in reasoning is established, we focus on enabling smaller models to learn *how to start correctly*. To this end, we introduce `QuestCoT`, a self-questioning guidance mechanism designed to teach models *how to start*. With `QuestCoT`, the model first generates a sub-question that initiates the reasoning chain, and then follows that path. Essentially, it identifies the most effective reasoning chains needed to answer the given question. A comparison of our proposed methodology, `QuestCoT` and Chain-of-Thought (CoT) is demonstrated in Figure 1.

We demonstrate the importance of self-questioning for initializing reasoning chains (`QuestCoT`) on several mathematical datasets involving multi-step word problems. Consistent performance improvements were observed for all smaller models (all within 7B parameters). Moreover, `QuestCoT` performs similarly to expert LLM guidance improving the quality of reasoning and outperforms the standard reasoning techniques of chain-of-thought (Wei et al., 2022, CoT) and sub-question decomposition approaches (Shridhar et al., 2022; Zhou et al., 2023, Subques).

## 2 RELATED WORK

It is possible to elicit reasoning abilities from LLMs through in-context learning, either by providing the model with intermediate steps Wei et al. (2022); Kojima et al. (2023); Yang et al. (2023); Wang et al. (2023b), or by decomposing the problem into smaller sub-problems Shridhar et al. (2022); Zhou et al. (2023) and solving them to reach the final answer. However, if the problem is misinterpreted, it can lead to a cascade of errors in subsequent steps.

To counter this, several techniques have been proposed to intervene and correct intermediate steps by providing feedback on their own generations, and eventually "self-correcting" their own generations Welleck et al. (2022); Madaan et al. (2023); Shridhar et al. (2023a). While the LLM's ability to revise its own generations may prove helpful in many cases, it sometimes leads to worse results in refinement, requiring a "rollback" to the previous output Shridhar et al. (2023a). To address this, Yao et al. (2023) introduces the Tree of Thoughts (ToT), which plans subsequent steps to solve a reasoning task Huang et al. (2022); Wang et al. (2023a;c). ToT conceptualizes the decision-making process as a series of heuristics based decisions. Through deliberate search, ToT explores different reasoning paths and self-reflects on its decision at each step. We, on the other hand, propose to get the first step right, thus reducing the cost of "finding" and "fixing" errors.

Previous work has also focused on understanding *when* to intervene and correct the errors. Saha et al. (2023) presented an approach based on Theory of Mind Kosinski (2023); Kadavath et al. (2022), where a teacher model intervenes in a student model only for harder questions by creating an implicit mental model of the student's understanding. In contrast, an alternative that avoids the need to backtrack and correct mistakes, thus saving time and effort, is to *start right*.

# 3 FIRST-STEP ADVANTAGE

In this section, we explore the impact of initiating the correct reasoning chain on the performance of smaller language models (LMs) in multi-step reasoning tasks. We address three key research questions:

1. Are smaller models capable of solving a reasoning task?
2. What is the importance of taking the correct first step in reasoning?
3. Can smaller models learn to take the correct first step on their own?

**Preliminaries** We start by defining a language model $\mathcal{M}$ that, given a question $q$, generates an answer $a$ through a reasoning process $r$. The model aims to find the most probable answer $a^*$ by maximizing $P(a \mid q)$.

In the standard setting, the model generates a sequence of tokens $y = (y_1, y_2, \ldots, y_T)$, where $y_t$ is the $t$-th token in the combined reasoning chain $r$ and answer $a$. The probability of generating $y$ is:

$$P(y \mid q) = \prod_{t=1}^{T} P(y_t \mid y_{<t}, q)$$

Keeping the above definition in mind, we explore the three research questions.

## 3.1 ARE SMALLER MODELS CAPABLE OF SOLVING A REASONING TASK?

**Hypothesis** *We hypothesize that the smaller models can solve a given task but are not confident enough to choose the correct reasoning chain.*

**Our Approach** To test the hypothesis, we take a question $q$, and use a smaller language model $\mathcal{M}$ to generate $K$ different reasoning chains $\{r^{(1)}, r^{(2)}, \ldots, r^{(K)}\}$ and corresponding answers $\{a^{(1)}, a^{(2)}, \ldots, a^{(K)}\}$. The probability of generating each reasoning chain is given by:

$$P(r^{(k)}, a^{(k)} \mid q) = \prod_{t=1}^{T} P(y_t^{(k)} \mid y_{<t}^{(k)}, q)$$

where $y_t^{(k)}$ is the $t$-th token in the $k$-th reasoning chain, and $y_{<t}^{(k)}$ denotes all previous tokens. We evaluate whether any of the $K$ generated answers $a^{(k)}$ matches the correct answer $a^*$.

**Experimental Design** We investigate the ability of smaller models to solve reasoning tasks by sampling their outputs multiple times and check whether a correct answer appears in any of them. We keep the sampling count $K$ to [1, 3, 5, 10, 15, 20, 25, 30, 35]. A temperature setting of 0.7 is used to generate diverse multiple samples. We compared the performance of the smaller model (Mistral-7B Jiang et al. (2023)) with the larger ones (LLaMA2-70B Touvron et al. (2023), and GPT-4 OpenAI (2023)) on the GSM8K dataset Cobbe et al. (2021) for mathematical reasoning. Our analyses were conducted on a test set of 1,319 samples using a 4-shot Chain-of-Thought (CoT) reasoning chain. The prompts used are listed in the Appendix (Figure 10).

**Results** Figure 2 illustrates the accuracy of whether an answer exists in one of the sampled outputs for different model sizes (ranging from Mistral 7B to LLaMA 70B to GPT-4, where the true size of GPT-4 is unknown but presumably large) when sampled multiple times. In a single sample, the performance gap between GPT-4 and Mistral 7B is nearly 50 points, indicating GPT-4's superior ability to select the correct reasoning chain in its initial sample.

**Question** The toothpaste in Anne's family's bathroom contains 105 grams. Anne's dad uses 3 grams at each brushing, her mom uses 2 grams, Anne and her brother use 1 gram each. Each member of the family brushes their teeth three times a day. How many days will it take for the toothpaste to run out?

**LLM-based first step guidance** Anne and her brother use 1 gram + 1 gram = «1+1=2»2 grams. Anne's dad uses 3 grams and Anne's mom uses 2 grams so together they use 2 grams + 3 grams + 2 grams = «3+2+2=7»7 grams/wash. Each day, the family uses 7 grams/wash * 3 washes/day = «7*3=21»21 grams. Using that amount daily, the toothpaste will last 105 grams / 21 grams/day = «105/21=5»5 days. The answer is 5.

Figure 3: LLM-based first step guidance is highlighted in yellow followed by model generation.

However, with 35 samples, this gap narrows to less than 10 points, suggesting that smaller models can answer correctly but struggle to consistently select the right chain in their first few attempts. Conversely, GPT-4's accuracy does not increase significantly with additional samples, suggesting greater consistency in generating and selecting the correct inference chain. This finding is consistent with Lyu et al. (2024), which suggests that scaling improves output consistency.

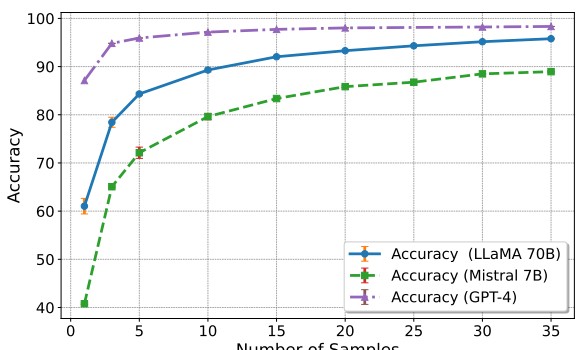

Figure 2: Accuracy (if an answer exists in one of the output chains) comparison on GSM8K data set between different sized models: Mistral 7B, LLaMA-70B, and GPT-4.

**Key Findings** We observed that smaller models can answer a reasoning question when sampled multiple times, but fail to select the correct reasoning chain on the first attempt.

### 3.2 IMPORTANCE OF STARTING RIGHT

Since smaller models are capable of solving a given question, why don't they choose the right reasoning chain on the first try? We believe that this is because the smaller models struggle to get the starting step right. An incorrect reasoning step can lead to error propagation due to the autoregressive nature of LMs. For example, Chain of Thought (CoT) Wei et al. (2022) is one of the most common reasoning strategies based on intermediate steps, and if an error occurs in the initial steps, it becomes difficult to correct due to its accumulation over steps. Let $\epsilon_t$ denote the error probability at step $t$. The cumulative error probability over $N$ steps is

$$P_{\text{error}}^{\text{CoT}} = 1 - \prod_{t=1}^{N}(1 - \epsilon_t)$$

An error in early steps significantly impacts the overall error probability due to the multiplicative effect.

**Hypothesis** *We hypothesize that the smaller models can solve a given task if they get the first step right.*

**Our Approach** To verify our hypothesis, we use a larger LLM, such as GPT-4 OpenAI (2023), to generate the first step of the solution $s_1$ for each question $q$. The smaller model $\mathcal{M}$ then generates the rest of the reasoning chain $r$ and the final answer $a$, conditioned on both $q$ and $s_1$:

$$P(a, r \mid q, s_1) = \prod_{t=1}^{T} P(y_t \mid y_{<t}, q, s_1)$$

We perform sanity checks to ensure that no answer is revealed in this step (detailed analysis in Section 4) and limit the first step to a maximum of one equation. Since the problem requires at least two to

Table 1: Accuracy comparison when the first step is provided by a larger LLM versus the baseline (no first step provided) for a smaller model. The best results are shown in **bold**. Note that when a weaker model provides guidance (LLaMA2-70B performance is worse than Phi3-mini), it hurts the performance (underlined).

| Model | GSM8K | | | | SVAMP | | | |
|---|---|---|---|---|---|---|---|---|
| | CoT | LLM Guidance | | | CoT | LLM Guidance | | |
| | No guidance | LLaMA2-70B | GPT-3.5 | GPT-4 | No guidance | LLaMA2-70B | GPT-3.5 | GPT-4 |
| Gemma-2B | 7.50 | 12.81 | 16.23 | **17.84** | 34.60 | 36.30 | 46.70 | **49.20** |
| Phi3-Mini-3.8B | 76.95 | 75.10 | 77.39 | **80.27** | 86.30 | 84.20 | 86.10 | **87.80** |
| LLaMA2-7B | 10.53 | 19.48 | 21.00 | **23.27** | 38.00 | 40.10 | 41.40 | **48.20** |
| OlMo-7B | 13.64 | 28.20 | 36.54 | **37.90** | 18.60 | 40.90 | 46.50 | **49.90** |
| Mistral-7B | 40.25 | 46.17 | 48.82 | **49.50** | 62.00 | 65.60 | 66.80 | **73.40** |
| Gemma-7B | 46.55 | 52.23 | 59.43 | **63.45** | 70.30 | 72.10 | 74.10 | **78.30** |

eight equations to solve, the first-step guidance does not lead directly to the answer but provides a solid starting point for the model. Figure 3 shows an example of LLM-based first-step guidance or $s_1$ (highlighted in yellow ).

**Experimental setup** We investigate whether providing *first-step* guidance can help smaller models get better results. We evaluate smaller models in the 2B - 7B range, namely Gemma-2B Team et al. (2024), Phi3-mini 3.8B Abdin et al. (2024), LLaMA2-7B Touvron et al. (2023), OlMo-7B Magnusson et al. (2023), Mistral-7B Jiang et al. (2023), and Gemma-7B Team et al. (2024). All the models are instruction-tuned versions except LLaMA2 and Mistral. For guidance coming from LLMs, we use LLaMA2-70B Touvron et al. (2023), GPT-3.5, and GPT-4 OpenAI (2023). We test our hypothesis on the test set of two datasets: GSM8K with 1319 samples and SVAMP Patel et al. (2021) with 1000 samples. Greedy sampling (temperature=0) was used for sampling and acc@1 accuracy is reported.

**Results** Table 1 demonstrates the usefulness of the first-step guidance provided by LLMs. The performance of the pre-trained models increases by more than 2-3X when a larger model such as GPT-4 is used for first-step guidance. For example, the performance of Gemma-2B Team et al. (2024) and LLaMA2-7B model Touvron et al. (2023) goes from $7.5 \rightarrow 17.8$ and $10.5 \rightarrow 23.2$, respectively, while for OlMo-7B it goes from $13.6 \rightarrow 37.9$ (an almost 3X jump). Performance increases monotonically with larger and more expert models providing first-step guidance (for Gemma-2B, performance increases from $7.5 \rightarrow 12.8$ with LLaMA-70B first-step guidance and to 16.2 with GPT-3.5). For the more expert models on the GSM8K task, Mistral-7B Jiang et al. (2023) gains almost +10 points ($40.25 \rightarrow 49.50$), Gemma-7B gains +17 points ($46.5 \rightarrow 63.4$), and Phi3-Mini Abdin et al. (2024) gains almost +4 points ($76.9 \rightarrow 80.2$). The trend is similar for the SVAMP dataset, where accuracy goes from $34.6 \rightarrow 49.2$ for Gemma and $38.0 \rightarrow 48.2$ for LLaMA2-7B with GPT-4 first-step guidance, and from $18.6 \rightarrow$ almost 50 for OlMo-7B. This shows the importance of the first-step guidance in arriving at the correct final answer.

**Key Findings** We observed that when the first step of the reasoning chain is provided to the smaller models as a starting point, the performance of the smaller models improves significantly.

### 3.3 CAN SMALLER MODELS LEARN TO GET THE FIRST STEP RIGHT?

Once we establish that smaller models can solve a given task if they get the first step right, can smaller models learn to get the first step right?

**Hypothesis** *Smaller models can learn to take the correct first step through self-questioning.*

**Our approach** We introduce `QuestCoT`, a self-questioning strategy where the model first generates a sub-question $s$ that guides the reasoning process.

$$s = \arg\max_{s'} P(s' \mid q)$$

---

**Question** Gene is sewing a quilt out of old souvenir t-shirts. He has one shirt from each vacation he has been on. Every shirt is its own quilt block. Each row is made of blocks from a different year of vacations. He goes on four vacations a year and has been vacationing since he was 23 years old. He is now 34. How many quilt blocks does he have in total?

**CoT** He has been on 4 vacations a year for 34 years so he has been on 4*34 = «4*34=136»136 vacations. Each vacation is a quilt block so he has 136 quilt blocks. Each quilt block is one shirt so he has 136 shirts. The answer is 136. ✗

**QuestCoT** How many years Gene has been on vacation? Gene has been on 34 - 23 = «34-23=11»11 years of vacations . Each row is made of 4 blocks, and she has 11 rows of blocks. So he has 11*4 = «11*4=44»44 blocks in total. The answer is 44. ✔

---

Figure 4: Example of a comparison between CoT reasoning and `QuestCoT`. `QuestCoT` first asks a question that helps to decide the first step and is highlighted in pink .

Then, the model generates the reasoning chain $r$ and the corresponding answer $a$ conditioned on the question $q$ and the sub-question $s$:

$$P(a, r \mid q, s) = \prod_{t=1}^{T} P(y_t \mid y_{<t}, q, s)$$

An example of our approach and its comparison to CoT is presented in Figure 4.

**Why `QuestCoT` works** Conditioning on the subquestion $s$ reduces the entropy of the model's output and focuses the model on the correct reasoning path.

By introducing the subquestion $s$, we increase the probability of generating the correct reasoning chain $r^*$ and answer $a^*$:

$$P(a^*, r^* \mid q) = \sum_{s} P(s \mid q) P(a^*, r^* \mid q, s)$$

Since the subquestion $s$ is designed to focus the model on the appropriate starting point, $P(a^*, r^* \mid q, s)$ is higher than without conditioning on the subquestion $s$.

Conditioning on $s$ reduces the cumulative error probability:

$$P_{\text{error}}^{\text{QuestCoT}} = 1 - (1 - \epsilon_1') \prod_{t=2}^{N} (1 - \epsilon_t')$$

where $\epsilon_1' < \epsilon_1$ and $\epsilon_t' \leq \epsilon_t$ for $t > 1$. Therefore, we have:

$$P_{\text{error}}^{\text{QuestCoT}} < P_{\text{error}}^{\text{CoT}}$$

In other words, conditioning on the subquestion $s$ increases the mutual information between the model's output and the correct answer:

$$I(a, r; a^* \mid q, s) = H(a, r \mid q, s) - H(a, r \mid a^*, q, s)$$

A lower entropy $H(a, r \mid q, s)$ implies that the model's predictions are more concentrated around the correct reasoning paths, increasing the likelihood of producing the correct answer.

**Experimental setup** We explore the effect of *starting right* on four multi-step mathematical data sets: GSM8K Cobbe et al. (2021), SVAMP Patel et al. (2021), ASDiv Miao et al. (2020), and MultiArith Roy & Roth (2015). GSM8K consists of grade-school math word problems with a test set of 1319 samples, requiring between two and eight steps to solve. SVAMP consists of 1000 samples of

Table 2: Accuracy comparison between the chain of thought (CoT) and `QuestCoT` . QuestCoT achieves the best results across all model sizes for various multi-step mathematical reasoning datasets.

| Model | Dataset | | | | | | | |
|---|---|---|---|---|---|---|---|---|
| | GSM8K | | SVAMP | | ASDiv | | MultiArith | |
| | CoT | QuestCoT | CoT | QuestCoT | CoT | QuestCoT | CoT | QuestCoT |
| Gemma-2B | 7.50 | **8.76** (↑ +1.1) | 34.60 | **35.00** (↑ +0.4) | 42.34 | **42.95** (↑ +0.6) | 17.77 | **18.88** (↑ +1.1) |
| Phi3-Mini-3.8B | 76.95 | **78.92** (↑ +2.0) | 86.30 | **88.40** (↑ +2.1) | 80.82 | **82.34** (↑ +1.5) | 98.83 | **99.44** (↑ +0.6) |
| LLaMA2-7B | 10.53 | **15.10** (↑ +4.5) | 38.00 | **41.10** (↑ +3.1) | **41.43** | 40.90 (↓ -0.5) | 25.55 | **28.88** (↑ +3.3) |
| OlMo-7B | 13.64 | **19.40** (↑ +5.8) | 18.60 | **27.20** (↑ +8.6) | 39.37 | **44.40** (↑ +5.0) | 20.00 | **27.22** (↑ +7.2) |
| Mistral-7B | 40.25 | **45.47** (↑ +5.2) | 62.01 | **65.15** (↑ +3.1) | 54.18 | **57.26** (↑ +3.0) | 61.66 | **65.55** (↑ +3.9) |
| Gemma-7B | 46.55 | **48.21** (↑ +1.6) | 70.30 | **71.40** (↑ +1.1) | 68.59 | **69.84** (↑ +1.2) | **79.44** | 78.22 (↓ -1.2) |
| LLaMA3-8B | 78.86 | **79.80** (↑ +1.0) | 83.70 | **84.89** (↑ +1.2) | 73.88 | **74.27** (↑ +0.4) | 97.77 | **98.33** (↑ +0.5) |

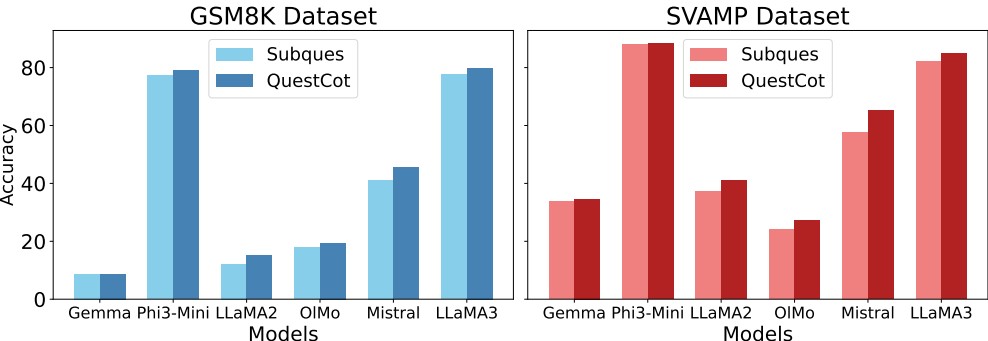

Figure 5: Accuracy comparison between Subques and `QuestCoT` on the GSM8K and SVAMP datasets. Gemma refers to Gemma-2B, Phi3-Mini is Phi3-mini-3.8B, and LLaMA2, OlMo, and Mistral are all 7B variants, while LLaMA3 is LLaMA3-8B.

math word problems designed to challenge systems that require reasoning beyond shallow approaches. ASDiv consists of 2,305 test samples of word problems that were constructed to have more lexical diversity than other datasets at the time. MultiArith is a dataset of 180 test samples published with the algorithmic solver for mathematical word problems.

We tested smaller models ranging from 2B to 8B parameters, starting with Gemma-2B, followed by Phi3-mini with 3.8B parameters, followed by Mistral-7B, LLaMA2-7B, OlMo-7B, and Gemma-7B with 7B parameters, and finally LLaMA3-8B with 8B parameters. We report the top-1 accuracy (`maj@1`) on the test sentences of both datasets. To compare CoT and `QuestCoT`, we used 4-shot prompting with prompts randomly selected from the test set. All models were evaluated using a greedy approach (temperature=0, top p=1). A comparison of prompts between CoT and `QuestCoT` can be found in the Appendix (Figure 11).

**Results** We test the effectiveness of `QuestCoT` against one of the most popular reasoning strategies: CoT. `QuestCoT` outperforms CoT on all four datasets for all models except LLaMA2-7B on ASDiv and Gemma-7B on MultiArith. Smaller models such as Gemma-2B and Phi-mini-3.8B gain between +0.5 and +2 points on all four datasets. We hypothesize that Gemma-2B's limited gains are due to its initial weak performance and undertraining, while Phi3-mini is already a very strong model with performance in the 80s and 90s, making further improvement difficult. Nevertheless, improvements are observed in both cases.

Performance improves significantly with the 7B models, with OlMo-7B showing the most gains (+6 on GSM8K, +9 on SVAMP, +5 on ASDiv, and +7 on MultiArith). This is followed by LLaMA2-7B and Mistral-7B, which show gains of +3 − 5 points, and Gemma-7B, which shows gains of +1 − 2

points. Similar to Phi3-mini, LLaMA3-8B's baseline performance is quite high, showing gains of +0.5 to +1 point.

**Key Takeaways** Smaller models improve their performance by learning to get the first step right by asking themselves how to start. This improvement is achieved with our proposed approach, `QuestCoT`.

## 4 ANALYSIS

**Can first-step guidance go beyond two-step problems?** Figure 6 illustrates the performance of the Mistral-7B model with and without first-step LLM guidance for different steps in the GSM8K dataset. For all steps (2 to 8), first-step guidance improves performance, suggesting that starting with a solid foundation can help reasoning over a longer context.

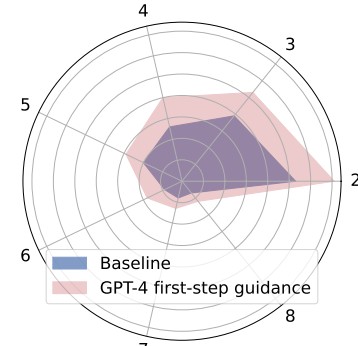

Figure 6: Accuracy comparison between baseline (no guidance) and LLM guidance (GPT-4) for the Mistral-7B model on the GSM8K dataset. 2-8 represents the number of steps required to solve the problem.

**Does the first step leak the final answer?** We investigate whether the performance gains from LLM guidance are due to LLMs leaking the answer to the smaller models. To verify this, we created a development set of 1000 samples from the GSM8K training set. By comparing the generated first-step answers with the final answers in the dataset, we found that in 999 out of 1000 samples, the answers did not match. Furthermore, our instructions to the LLMs specified that they could only generate the first step, corresponding to the first step in the inference chain with only the first equation, and could not reveal the final answer. This strategy was applied consistently across all data sets. Since each question required at least 2-8 steps to solve, we are confident that the final answer was not revealed. Furthermore, if the approach relied on revealing the final answer, the `QuestCoT` approach would not have been effective in the prompt style at all.

**Comparing CoT, Subques and `QuestCoT`** Figure 7 presents a Venn diagram illustrating when different strategies correctly solve problems on the GSM8K dataset using the Phi-3 mini-model. `QuestCoT` has more overlaps with both CoT and Subques (82 and 95, respectively) than the overlaps between CoT and Subques (71). This suggests that while CoT and Subques each have unique cases where they are successful, `QuestCoT` captures the strengths of both strategies, resulting in better overall performance.

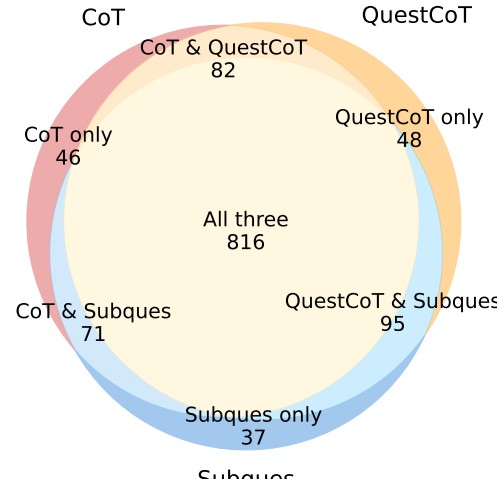

Figure 7: Venn diagram to show when different strategies got the solutions right.

**What if subquestions are included at each step?** The subquestion that guides the model on *how to start* can be applied to any reasoning step in the chain to guide that specific step. This approach is similar to subquestion decomposition (Shridhar et al., 2022; Zhou et al., 2023, Subques), where a complex reasoning problem is first broken down into simpler problems that are then solved sequentially. Figure 5 shows a comparison between Subques and `QuestCoT` over different models on two datasets: GSM8K and SVAMP. `QuestCoT` shows higher accuracy across all models on both datasets while incurring lower token costs. Through manual inspection of over 100 samples, we found that introducing additional questions increases the likelihood of errors propagating through the inference chain. In contrast,

---

**Unnecessary calculations**

**Question** Roger goes to the store to buy some coffee. The normal brand of coffee he buys cost $5 per pound. He had to buy a more expensive brand that cost 20% more since his favorite brand was sold out. He decides to buy a week's worth of coffee and he uses 1 pound of coffee per day. He also decided to buy himself a donut for $2. How much did everything cost?

**CoT** Roger buys 7 pounds of coffee at $5 per pound, so the cost of coffee is 7 * $5 $«7*5=35»35. The more expensive brand of coffee costs 20% more, so it costs $5 * 1.20 = $«5*1.20=6»6 per pound. The cost of the more expensive coffee is 7 * $6 = $«7*6=42»42. Roger also buys a donut for $2. The total cost of everything is $42 + $35 + $2 = $«42+35+2=79»79. Final Answer: 79 ✗

**QuestCoT** What is the cost of the more expensive brand of coffee? The more expensive brand costs 20% more than the normal brand, so it costs $5 * 20/100 = $«5*20/100=1»1 more per pound. So, the more expensive brand of coffee costs $5 + $1 = $«5+1=6»6 per pound. Roger buys a week's worth of coffee, which is 7 days, and he uses 1 pound of coffee per day. So, he buys 7 pounds of the more expensive coffee. The total cost of the coffee is 7 pounds * $6/pound = $«7*6=42»42. He also buys a donut for $2. So, the total cost of everything is $42 (coffee) + $2 (donut) = $«42+2=44»44. Final Answer: 44 ✓

---

**Real world knowledge**

**Question** Twenty dozen cups cost $1200 less than the total cost of half a dozen plates sold at $6000 each. Calculate the total cost of buying each cup.

**CoT** Let's find the total cost of half a dozen plates: 6000 * 0.5 = $3000 Now, let's find the total cost of twenty dozen cups: $3000 + $1200 = $4200. Since there are 12 cups in a dozen, there are 20 * 12 = «20*12=240»240 cups in twenty dozen. The cost of each cup: $4200 / 240 = $17.50. Final Answer: 17.50 ✗

**QuestCoT** How many cups are in 20 dozen? 1 dozen is equal to 12 cups, so 20 dozen is equal to 20 * 12 = «20*12=240»240 cups. Half a dozen plates is equal to 0.5 * 12 = «0.5*12=6»6 plates. The total cost of half a dozen plates is 6 * $6000 = $«6*6000=36000»36000. The total cost of buying 20 dozen cups is $1200 less than the cost of half a dozen plates, so it is $36000 - $1200 = $«36000-1200=34800»34800. The total cost of buying 20 dozen cups is $34800, so the cost of each cup is $34800 / 240 = $«34800/240=145»145. Final Answer: 145 ✓

---

**Context understanding**

**Question** You can buy a movie super ticket for $20 that includes rights to watch any movie and you get one popcorn with a soda for only $1 extra. Or you can buy a movie ticket for $12 and buy one popcorn and a soda at regular price. What's the regular price of popcorn if the soda costs $3 and you didn't buy the super ticket and ended up saving $2?

**CoT** Let's denote the regular price of popcorn as P. If you buy the regular ticket and popcorn and soda, the total cost would be:$12 (movie ticket) + P (popcorn) + $3 (soda) = $12 + P + $3. If you buy the super ticket and popcorn with soda, the total cost would be:$20 (super ticket) + $1 (popcorn) + $3 (soda) = $20 + $1 + $3. You saved $2 by not buying the super ticket, so the cost of the regular ticket and popcorn and soda is $2 more than the cost of the super ticket and popcorn and soda:$12 + P + $3 = $20 + $1 + $3 + $2. Now, let's solve for P:$12 + P + $3 = $20 + $1 + $3 + $2. $12 + P + $3 = $26. P + $15 = $26. P = $26 -$15 = $3. Final Answer: 3 ✗

**QuestCoT** What is the total cost of buying a ticket, popcorn and soda? Let's denote the regular price of popcorn as P. The total cost of buying a regular ticket, popcorn, and soda is $12 + P + $3 = $15 + P. The total cost of buying a super ticket, popcorn, and soda is $20 + $1 = $21. Since you saved $2 by not buying the super ticket, the cost of the regular ticket, popcorn, and soda is $21 - $2 = $19. Therefore, $15 + P = $19. Solving for P, we get P = $19 - $15 = $4. Final Answer: 4 ✓

---

Figure 8: Example cases associated with different error types and how `QuestCoT` assists in getting the correct reasoning chain.

`QuestCoT` guidance avoids this by focusing solely on *how to start*. These errors can result from incorrect decomposition, incorrect reasoning about the decomposed problem, or inconsistencies throughout the inference chain.

**Can `QuestCoT` work with even smaller models?** We tested our approach on the OlMo-1B model, which has 1 billion parameters, and found that it was not well equipped to understand the instructions or to generate a reasoning chain starting with an initial question (a necessary condition for `QuestCoT`). On the GSM8K dataset, the CoT performance was 3%, and `QuestCoT` performed comparably at 3.5%, with the outputs not looking significantly different. As a result, we did not observe any statistically

significant improvements. We suspect that because the OlMo-1B model's CoT abilities are quite limited in mathematical reasoning tasks, it cannot leverage any advantages from `QuestCoT`.

## 5  DEEPER EXPLORATION OF WHY `QuestCoT` WORKS

Understanding why one technique outperforms another can be quite challenging. To address this, we examined instances where `QuestCoT` was successful and conducted a detailed analysis. We identified situations where CoT failed and broadly categorized these errors. Here are some error types where `QuestCoT` was beneficial:

**Unnecessary calculations**    A common mistake CoT makes is performing unnecessary calculations on the numbers in the statement. These numbers may be completely irrelevant to the problem, or they may need to be used in a different way than the model uses them. `QuestCoT` helps to correct these errors by initiating the reasoning process with an appropriate question. An example of an unnecessary calculation is given in Figure 8 with the main error shown in red, where there was no need to calculate the total cost of coffee for the entire week at the old price.

**Real-world knowledge**    Often, the first step in CoT seems somewhat "rushed," focusing on quickly manipulating numbers without considering real-world facts or knowledge. In these cases, the model demonstrates its understanding of these facts and knowledge in the subsequent steps but cannot elicit it immediately in the first step. This suggests that encouraging the model to think more deliberately in the first step (e.g., by allowing it to consider what needs to be done before it starts reasoning) may remedy this problem. These scenarios illustrate the effectiveness of `QuestCoT`. An example is shown in Figure 8, where the model fails to convert "half a dozen" to 6, and instead continues its calculations with 0.5 (as shown in blue). Although the model demonstrated its understanding of "dozen" later in the problem, since it started incorrectly, it was unable to correct the chain later.

**Context understanding**    With CoT, the model often confuses or misses the context in the problem statement and makes incorrect initial assumptions that are difficult to recover from in later steps. For example, in Figure 8 we can see that despite following a fairly elaborate reasoning template of variable assumptions, the CoT reasoning misses the fact that the price of the Super Ticket already includes the price of the popcorn. The incorrect assumption is highlighted in the response.

**Other errors**    Other errors we have observed include that `QuestCoT` may be better at handling direct numeric computations and understanding the simple arithmetic required by the problems. In contrast, CoT may deviate or fail to capture the essential computational aspects of the query. In addition, CoT sometimes takes more steps than necessary, resulting in an incorrect final solution.

## 6  CONCLUSION

We find that smaller models sometimes struggle with taking the correct first step, but their performance increases significantly once this step is corrected. We demonstrated this by using LLMs to guide smaller models to take the correct first step, helping them to establish the correct reasoning chain. To facilitate this for smaller models, we propose `QuestCoT`, which uses initial question-based guidance to improve their reasoning themselves without any guidance. We demonstrate the effectiveness of our approach on four multi-step mathematical reasoning datasets using different open-source small models.

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

## A APPENDIX

Below is a math word problem that requires multiple steps to solve it. Your job is to only provide the first step of the solution and not to reveal the final answer. The first step consists of only one equation in it.

### Input: Thomas is training at the gym to prepare for a competition. He trained for 5 hours every day for a month (30 days). If he continues to train for the next 12 days, how many hours will he spend on training in total?

### Response: Total hours for first month=5hours/day×30days

Figure 9: Instructions to generate first step by LLM. The model-generated output is presented in green.

Below is an instruction that describes a task, paired with an ### Input that provides further context. Write a ### Response that appropriately completes the request.

### Instruction: Solve the given math problem step by step, and put your final answer after 'Final answer:'.

### Input: Thomas is training at the gym to prepare for a competition. He trained for 5 hours every day for a month (30 days). If he continues to train for the next 12 days, how many hours will he spend on training in total?

### Response: In total Thomas would train on 30 + 12 = «30+12=42»42 days. Thomas trained 5 hours every day, which would bring us to 42 * 5 = «42*5=210»210 hours of training in total. Final Answer: 210 <eot_id>

**[Similar 3 more examples randomly sampled from the training set]**

### Input: Nina made a two-layer cake and a dozen cupcakes for her friend's birthday party. Each layer of cake takes the same amount of sugar to make as a dozen cupcakes. Nina used 720 grams of sugar to bake everything. How many grams of sugar are in one cupcake?

### Response: Model generated response ..

Figure 10: Four-shot CoT demonstration.

756
757
758
759
760
761
762
763
764
765
766
767
768
769
770
771
772
773

774 Below is an instruction that describes a task, paired with an ### Input that provides further context. Write a ###
775 Response that appropriately completes the request.

776 ### Instruction: Solve the given math problem step by step, and put your final answer after 'Final an-
777 swer:'.

779 ### Input: Thomas is training at the gym to prepare for a competition. He trained for 5 hours every
780 day for a month (30 days). If he continues to train for the next 12 days, how many hours will he spend on
781 training in total?

782 ### Response: **How many days will Thomas train in total?** In total Thomas would train on 30 + 12
783 = «30+12=42»42 days. Thomas trained 5 hours every day, which would bring us to 42 * 5 = «42*5=210»210
784 hours of training in total. Final Answer: 210 <eot_id>

785 **[Similar 3 more examples randomly sampled from the training set]**

787 ### Input: TNina made a two-layer cake and a dozen cupcakes for her friend's birthday party. Each layer of cake
788 takes the same amount of sugar to make as a dozen cupcakes. Nina used 720 grams of sugar to bake everything.
789 How many grams of sugar are in one cupcake?

790 ### Response: Model generated response ..

Figure 11: Four-shot `QuestCoT` demonstration. The only difference from CoT is **underlined**.
