# OpenReview forum: "First-Step Advantage: Importance of Starting Right in Multi-Step Math Reasoning"
_ICLR.cc/2025/Conference — ICLR 2025 Conference Withdrawn Submission_

### Official Review · Reviewer_VP12 · 2024-10-27

**Soundness:** 3
**Presentation:** 3
**Contribution:** 2
**Rating:** 3
**Confidence:** 4

**Summary:**

This work focuses on analyzing and improving mathematical reasoning for relatively small language models.
The authors experimentally demonstrate the importance of the initial step in the multi-step reasoning process and also show the limitation of small language models.
Moreover, the paper introduces a novel approach called QuestCoT (Questioning Chain of Thought), which enables smaller models to self-generate a preliminary question that guides the subsequent reasoning chain.

**Strengths:**

1. This work validate the importance of the initial step in the multi-step reasoning.
2. The authors provide experimental results to reveal that small language models can achieve the correct final results with multiple sampling, but fail to accurately find out the correct one.
3. This work QuestCoT to improve the reasoning performance of small language models by improving the generation quality of the initial step.

**Weaknesses:**

The main limitation lies in the novelty of this work.
For the three key research questions explored in this work, the first two about the capability of small language models have been well studied in [1].

[1] Common 7B Language Models Already Possess Strong Math Capabilities.

**Questions:**

1. As this work aims to emphasize the importance of the first step, is there any evidence to support that the first step is more important than other intermediate steps?

---

### Official Review · Reviewer_LDBC · 2024-10-31

**Soundness:** 3
**Presentation:** 3
**Contribution:** 3
**Rating:** 5
**Confidence:** 5

**Summary:**

Large Language Models (LLMs) have the capability to solve complex reasoning tasks by generating each intermediate reasoning step. Previous works show that model accuracy improves significantly when multiple reasoning chains are generated, indicating that the model understands how to answer the given problem. However, the model often struggles to select the correct initial chain and when it starts on an incorrect path it is difficult to fix it due to its autoregressive nature of decoding. Therefore, the author in this work answers three main questions:
1. Are smaller models capable of solving a reasoning task?

By sampling multiple reasoning chains while keeping a high temperature, they show that smaller models can reach the correct answer but struggle to consistently select the right chain in their first few attempts.

2. What is the importance of taking the correct first step in reasoning?

Here the author investigates the change in performance when the first step is provided as input along with the question. They observe a significant improvement in the accuracy. Therefore choosing the right first step is very important.

3. Can smaller models learn to take the correct first step on their own?

Here they introduce their novel framework called QuestCOT, where the model self questions about the first step before starting to answer the question. When asking the subquestion it helps the model focus on the correct reasoning path, as the model’s predictions are more concentrated around the correct reasoning path, increasing the likelihood of producing the correct answer.

The author then compared QuestCOT with COT on GSM8K and SVAMP datasets, showing its increased performance.

**Strengths:**

1. The paper proposes a new framework to increase model robustness to solve mathematical word problems.
2. The author clearly demonstrates the problems with current SLMs and shows how providing the first step will boost model performance.
3. Extensive experiments to answer the three questions stated in Section 3
4. Great analysis of the results, where the model talks about the effect of first-step guidance on a longer reasoning chain, does the first step leak the final answer? Etc

**Weaknesses:**

1. It is unclear how the questions are generated for each question, the prompt used to generate them, and how the quality of questions asked affects the performance of the model. An ablation study on the subquestions asked to the model can help better understand QuestCOT
2. In Section 3, the author talks about how “introducing additional questions increases the likelihood of errors propagating through the inference chain”. Can the author provide more evidence on why error increases when subquestions are asked at every step vs. only on the first step
3. Running more baselines: Comparison with more existing methods such as Self-consistency[1], Tree of thought [2] etc and more difficult datasets such as MATH, MATHBENCH etc,  will help prove the robustness of QuestCOT.

Reference:

[1] : Wang, Xuezhi, et al. "Self-consistency improves chain of thought reasoning in language models." arXiv preprint arXiv:2203.11171 (2022).

[2] : Yao, Shunyu, et al. "Tree of thoughts: Deliberate problem solving with large language models." Advances in Neural Information Processing Systems 36 (2024).

**Questions:**

Address the weakness pointed out in the above section

---

### Official Review · Reviewer_Um1G · 2024-11-03

**Soundness:** 2
**Presentation:** 3
**Contribution:** 2
**Rating:** 3
**Confidence:** 4

**Summary:**

The authors note that smaller models, when corrected, can handle tasks better, which is demonstrated by using larger models to guide them, resulting in improved performance (up to +24 points on the GSM8K dataset for 7B models). They propose QuestCoT, where a smaller model first asks how to start before proceeding with reasoning. The method shows performance gains across various mathematical reasoning datasets (up to +6 points on GSM8K, +9 on SVAMP, +5 on ASDiv, and +7 on MultiArith).

**Strengths:**

- The paper is well-written with clear and fluent prose. Additionally, the figures and tables are well-designed and effectively present the data.
- The key finding that smaller models can significantly improve their performance when corrected by larger models is insightful and empirically investigated well.

**Weaknesses:**

- The novelty of the QuestCoT method is somewhat lacking. It seems more like a new template for chain-of-thought reasoning presented in a question-and-answer-in-first-step format rather than a fundamentally innovative approach.
- The "Why QuestCoT works" part in Section 3.3 is confusing, especially when it is associated with the "What if subquestions are included a teach step" part in Section 4. If the analysis in "Why QuestCoT works" makes sense, we could also draw a conclusion that Subques would be better than QuestCoT.This contradicts the results in "What if subquestions are included a teach step". This paper lacks analysis and explanation for this discrepancy.

**Questions:**

My main concerns have already been expressed in the "weakness" section.

---

### Official Review · Reviewer_bazb · 2024-11-04

**Soundness:** 3
**Presentation:** 3
**Contribution:** 2
**Rating:** 3
**Confidence:** 4

**Summary:**

This paper introduces QuestCoT, where the model is instructed to first ask itself how to start, then start the chain-of-thought reasoning by its self-generated guidance question. The QuestCoT approach is motivated by the observation that the first step correctness is important for the model to achieve the correct final answer. The authors further show that if there is a stronger larger model to guide the first step, the small model would consequently reason better. The authors also provide a list of analyses about why QuestCoT works.

**Strengths:**

The authors clearly articulated why the first step matters, and give a detailed performance analysis of Quest CoT

- Clear articulation of why the first step matters: the authors use experimental results and concrete examples to show that the first step correctness is important to final answer accuracy.
- Additional analysis: the authors use fair followup analysis to discuss more details like data leakage, comparisons between different types of CoT, and if QuestCoT work with smaller models.

**Weaknesses:**

While the authors take inspiring steps towards strengthening CoT, I generally feel the experiments are insufficient to support an ICLR quality paper. Specifically:

- The GSM8K performance of Gemma 2B in this paper Table 2 is only 7.5, while on its official report is 17.7, the gap is 10.2. The Mistral 7B performance in Table 2 is 45.4, while in Mistral blog is 52.1, the gap is 6.9. Such large gap is in generally questionable. Given that Gemma and Mistral are well-established models whose performance is validated by a large portion of the community, I wonder why the authors’ numbers are so different than their official reports.
- This paper only considers GSM8K, SVAMP, ASDiv and MultiArith, all of them are quite old. The field have moved on, and the authors need to use stronger models (like Qwen 2, DeepSeek) and harder and broader datasets (MMLU-Pro, GPQA .etc) to demonstrate the effectiveness of their methods — simply use GSM8K-sytle benchmarks is insufficient.

**Questions:**

I wonder why the Gemma 2B and Mistral 7B GSM8K performance is significantly below the officially reported numbers. This may need more clarification to make the results more credible

---

### Note · Authors · 2024-11-13

**Comment:**

We thank all the reviewers for their feedback.

**Withdrawal Confirmation:**

I have read and agree with the venue's withdrawal policy on behalf of myself and my co-authors.